# Self-supervised contrastive learning improves machine learning discrimination of full thickness macular holes from epiretinal membranes in retinal OCT scans

Timothy William Wheeler[1]*, Kaitlyn Hunter[2], Patricia Anne Garcia[2], Henry Li[2], Andrew Clark Thomson[2], Allan Hunter[2], Courosh Mehanian[1,3]

**1** Department of Bioengineering, University of Oregon, Eugene, Oregon, United States of America, **2** Oregon Eye Consultants, Eugene, Oregon, United States of America, **3** Global Health Labs, Bellevue, Washington, United States of America

* twheele3@uoregon.edu

**Data Availability Statement:** The data that supports the findings of this study are available in Harvard Dataverse at https://doi.org/10.7910/DVN/

## Abstract

There is a growing interest in using computer-assisted models for the detection of macular conditions using optical coherence tomography (OCT) data. As the quantity of clinical scan data of specific conditions is limited, these models are typically developed by fine-tuning a generalized network to classify specific macular conditions of interest. Full thickness macular holes (FTMH) present a condition requiring urgent surgical repair to prevent vision loss. Other works on automated FTMH classification have tended to use supervised ImageNet pre-trained networks with good results but leave room for improvement. In this paper, we develop a model for FTMH classification using OCT B-scans around the central foveal region to pre-train a naïve network using contrastive self-supervised learning. We found that self-supervised pre-trained networks outperform ImageNet pre-trained networks despite a small training set size (284 eyes total, 51 FTMH+ eyes, 3 B-scans from each eye). On three replicate data splits, 3D spatial contrast pre-training yields a model with an average F1-score of 1.0 on holdout data (50 eyes total, 10 FTMH+), compared to an average F1-score of 0.831 for FTMH detection by ImageNet pre-trained models. These results demonstrate that even limited data may be applied toward self-supervised pre-training to substantially improve performance for FTMH classification, indicating applicability toward other OCT-based problems.

## Author summary

Full thickness macular holes (FTMH) are a sight-threatening condition that involves the fovea, the area of the retina of the eye involved in central vision. Timely diagnosis is paramount because of the risk of permanent vision loss with delayed surgical correction. In clinical practice, FTMH are commonly diagnosed with the aid of optical coherence tomography (OCT) images of the fovea. However, certain conditions such as pseudoholes

X3L0XD. All code for model training and analysis are available in GitHub at https://github.com/twheele3/rascl.

**Funding:** The author(s) received no specific funding for this work.

**Competing interests:** The authors have declared that no competing interests exist.

and epiretinal membranes may complicate the diagnosis of full thickness macular holes on imaging. Here, we employ artificial intelligence and present a machine-learning model that distinguishes full thickness macular hole from conditions that may present similarly upon image review. Despite training our model with a smaller data set, it outperformed traditional models previously seen in other works. We provide a strong framework for a self-supervised pre-trained model that can accurately distinguish full thickness macular holes from epiretinal membranes and pseudoholes. Overall, our study provides evidence of the benefit and efficacy of utilizing artificial intelligence for this image classification task.

## Introduction

Optical coherence tomography (OCT) allows detailed anatomical imaging via high resolution scans through several hundred microns of retinal tissue layers [1]. The macula is a 5.5 mm wide area of the posterior retina and includes a smaller 1.5 mm wide area known as the fovea which correlates with the eye's central and highest acuity vision [2,3]. OCT scanning of the macula generates a wealth of data on possible pathologies present within the retina, but to extract clinically relevant information requires subspecialty review. This imposes an increased workload on ophthalmologists, taking time away from higher level decision-making and patient care.

One condition of interest is full thickness macular holes (FTMH), wherein all layers of the neurosensory retina (between the internal limiting membrane and the retinal pigment epithelium (RPE)) in the fovea are disrupted [4]. The vitreous is a clear gel substance which fills the eye's posterior chamber and has attachments to the macula. Macular holes often develop from vitreous traction at the vitreofoveal interface and are most commonly idiopathic but can result from traumatic injury to the eye or secondary to other ocular pathologies [4,5]. Gass and Johnson were the first to suggest that this phenomenon occurs due to focal shrinking of the vitreous which results in traction on the fovea [4,5]. In Gass' original classification [4,6], the natural progression of macular holes proceeds through stages associated with worsening visual acuity: foveolar detachment in stage 1, formation of a foveal hole with associated subretinal fluid and cystoid macular edema in stage 2, formation of an FTMH in stage 3, followed by an FTMH with a complete posterior vitreous detachment from the vitreofoveal interface in stage 4. Due to involvement of the fovea, this results in impairment of central visual acuity [4].

A confounding comorbidity to FTMH is epiretinal membrane (ERM), in which a fibrotic contractile plaque grows over the retina that may result in anatomic changes similar in clinical appearance to FTMH (*e.g.*, pseudoholes or lamellar holes). ERMs are sometimes associated with macular edema which further distorts retinal anatomy, complicating FTMH diagnosis [7]. Furthermore, FTMHs and ERMs are both common conditions referred to retina practices and are both treated with pars plana vitrectomy (PPV), where the vitreous is surgically removed, and internal limiting membrane peel (ILMx), which involves removing the most superficial layer along the macula. These procedures eliminate all fovea deformation force(s) so that the retina can re-approximate to a more normal anatomical appearance and function. The clinical demand and common treatments for ERM and FTMH highlight the importance of machine learning classification.

Although the appropriate timing of ILMx surgery for FTMH is debated, there is evidence that extended delay in surgical repair heralds impaired vision [8]. In contrast, many ERM patients can be managed with conservative, non-surgical medical care [7]. The need for

accurate and timely diagnosis of FTMH can be critical for restoring vision, prompting a need for timely assessment of OCT scans. Automated classification can significantly reduce referral processing time allowing expeditious patient triage and care [9,10].

Deep learning has emerged as a promising avenue of research for creating automated expert reader models to assess medical scans in-line with the acquisition process. Briefly, deep learning refers to (artificial) neural networks that are many layers deep [11]. Neural networks, in turn, are defined as layers of nonlinear signal processing units (artificial neurons) connected by weights (artificial synapses). The effect of training a neural network is to adjust the weights such that the system approximates a function mapping a given input to a desired output. A multi-layered network of nonlinear processing units can in theory approximate any function to arbitrary precision with sufficient training data [12]. Convolutional neural networks (CNN) are particularly well-suited to image recognition tasks because the shared-weight architecture of convolutional kernels learns translation-covariant visual features automatically [13].

Developing an accurate deep learning model requires large quantities of well-annotated ground truth data, and a training regimen that allows the model to learn discriminative features for the problem at hand. A frequent issue arising in training deep learning models with medical data is low dataset size. This makes it impractical to train a naïve network from scratch and has typically been circumvented by fine-tuning a pre-trained network (transfer learning). This can also be problematic, as pre-training is usually based on images from a non-medical domain, typically ImageNet, which means that features extracted by the network may not be relevant to the specific nuances of the tissue and imaging method.

Self-supervised learning is an alternative to transfer learning for handling low data volume. In this approach, a suitable image representation is learned by performing a pretext task on unlabeled data, which is usually available in larger quantities than labeled data. Various approaches to self-supervised learning have been introduced, many involving an encoder, which is a feedforward neural network that represents an image by a vector (also known as an embedding). Contrastive learning pushes positive pairs closer and negative pairs farther apart in the embedding space. A primary example of this contrastive learning approach is SimCLR [14,15].

The last several years have seen an explosion of deep learning models applied to ophthalmic clinical technologies including OCT and fundus imaging. These applications may be divided into broad areas of classification/diagnosis [16–26], segmentation [27–34], image quality [35], and demographics prediction [36]. The current ophthalmic deep learning models focus primarily on diabetic retinopathy, age-related macular degeneration, retinopathy of prematurity, and glaucoma [9,37,38].

Deep learning for OCT image analysis of FTMH has also received attention lately, with models for classification [24,25], segmentation [31–34], and prognosis of success for FTMH corrective surgery [26,39,40]. A review is also available [41]. Owing to the paucity of labeled FTMH data, the majority of these models are pre-trained on ImageNet, and subsequently fine-tuned on a small amount of labeled FTMH OCT images. This transfer learning scheme was adopted in developing the classification model by Pace et al. [24], which achieves 95% accuracy in distinguishing between normal, Drusen, and FTMH images. Carvalho et al. [25], demonstrated an accuracy of 90.6% for FTMH identification, although modeling specifics are not provided. While these results are good, there is room for improvement.

We employ a pre-training method based on a variant of SimCLR that leverages 3D information, and which is tailored to small datasets with multi-slice imaging modalities such as OCT, described here as Random Slice Contrastive Learning (RaSCL). Our model achieves robust feature recognition in OCT scans for assessing FTMH that can outperform traditional ImageNet pre-trained models.

## Materials and methods

### Patient data

This Institutional Review Board (IRB) approved single center retrospective study reviews for OCT images of patients prior to PPV and ILMx surgery. Three researchers reviewed the Oregon Eye Consultants, LLC database using the ILMx procedure code (67042). Data used in this study was acquired from patients prior to surgical intervention [42]. OCT images that featured idiopathic FTMH (Gass stage 3 or 4) or ERMs (without macular holes) were included in the working dataset. Patients with non-idiopathic macular holes (traumatic, pseudohole), history of ocular trauma, amblyopia, recent ocular surgery (within three months), severe ocular myopia, diabetic macular retinopathy, and retinal pathology associated with systemic conditions (uncontrolled hypertension) were excluded. Variables including age, gender, lens status (pre- or post- cataract surgery), pre- and post- operative visual acuity, medical and ophthalmic comorbidities, and surgical history for patients who met the inclusion and exclusion criteria were documented and pre-operative OCT B-scans were exported for development of the model. OCT images were reviewed by three trained readers to confirm diagnoses (FTMH (Gass stage 3 or 4) and ERM) and assess image quality.

A B-scan is a horizontal linear OCT scan, producing a 2D image of a macular tissue section. Typically, multiple uniformly spaced B-scans are acquired, with the central scan going through the fovea. B-scans were individually labeled as having FTMH by consensus of three expert readers. Following diagnosis confirmation, the central B-scan for each patient was determined and documented by the same trained readers. The B-scans were exported from Heidelberg Spectralis OCT instruments (Heidelberg, Germany) as TIFF files at $496 \times 512$ resolution, with pixel size $3.87 \times 11.38$ μm at the retina. Macular OCT scan protocols available included B-scans at either 243 μm or 121 μm apart, and both protocol types were included.

### Test splits

Test sets for model evaluation were assembled from 15% of total eyes, comprising 10 FTMH and 40 control eyes, using only the central B-scan to ensure consistency between eyes with differing numbers of B-scans that show evidence of FTMH. Three random split replicates with disjoint test sets were generated for statistical validation. Within each replicate, self-supervised pre-training and fine-tuning were performed using only the training split. For each replicate, training sets were further divided into 8-fold cross-validation splits.

### Data stratification

Data was stratified based primarily on diagnosis (FTMH or ERM). Secondary features of age, sex, and pre-operative vision were used to stratify data by optimizing the mean and standard deviation of each subset relative to the overall dataset mean and standard deviation for each feature (treating sex as ordinal).

### Preprocessing

B-scans were cropped and resized to $224 \times 224$ resolution, and then augmented with random noise, brightness, contrast, cropping, and horizontal flips. Additive noise was generated by sampling a normal distribution with randomly drawn parameters $\mu \in [-0.1, 0.15]$ and $\sigma^2 \in [0, 0.2]$, cropping values to the interval $[0, 1]$, to mimic noise patterns typical of B-scans. Random crops comprised 50–100% of the original scan area.

## Architecture

The model architecture consists of a CNN followed by a multi-layered perceptron (MLP) based on the SimCLR approach [14,15]. The CNN uses a naïve ResNet50 framework at standard width [43]. The MLP comprises three fully connected layers of 512 nodes each. The final layer is used as a projection head during pre-training, which is fed to a classifier head during fine-tuning. Models were trained for 800 epochs during pre-training and 500 epochs during fine-tuning and were validated every 10 epochs. An 8-fold training set split was used for cross-validation training and unweighted ensemble averaging.

Training was performed on a workstation with an NVIDIA RTX A6000 GPU, a 24-core Intel Xeon W 3345 CPU, under Ubuntu 20.04, using Python 3.8.10 and TensorFlow 2.9.0. Taken together, pre-training and fine-tuning ran for 7.5 hours per ensemble replicate in this computational framework.

## Self-supervised pre-training

Pre-training was performed using contrastive self-supervised learning. Image slices from the same eye constituted positive pairs, while images from unrelated eyes constituted negative pairs. In more detail, for each batch, a B-scan was randomly selected from an eye, then a neighboring slice at a distance up to 2 slices away on either side was randomly selected (uniformly) as a positive pair. Scans between unrelated eyes constituted negative pairs (Fig 1). We refer to our scheme as 3D spatial contrast or Random Slice Contrastive Learning (RaSCL), which is similar in spirit to the SimCLR adaptation in Gomariz, *et al*. [34]. However, in the latter, positive pair slice distance is distributed normally: $d \sim \mathcal{N}(0, 0.25\mu m)$, and since their B-scans are separated by $111\mu m$, their scheme is effectively the same as standard SimCLR.

The model transferred for fine-tuning was selected based on the training epoch with maximal accuracy and minimal loss, *i.e.*, min(loss/accuracy), on the validation set. Pre-training data included three B-scans per eye centered on the fovea.

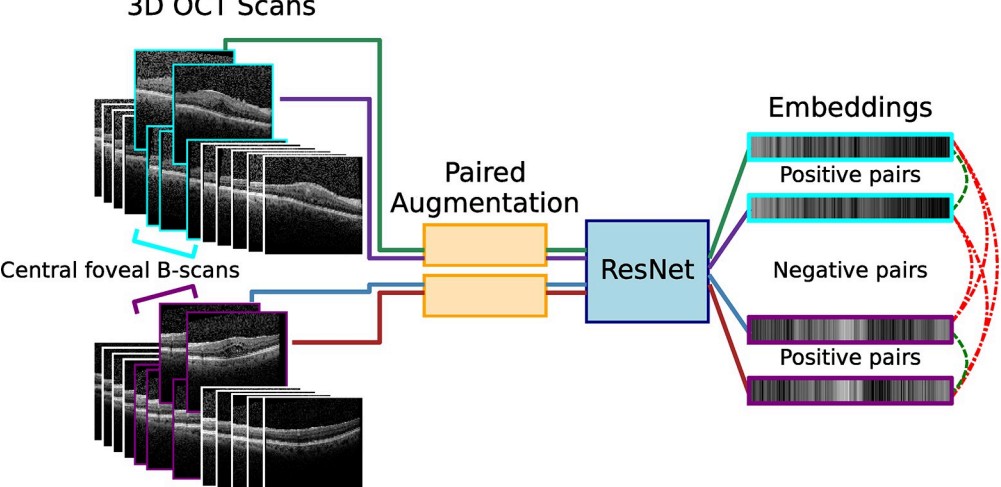

**Fig 1. An overview of the Random Slice Contrastive Learning (RaSCL) method.** Pairs of scans from the same eye and a different eye are randomly chosen from around the central foveal B-scan within a selection margin. Image pairs are then augmented independently and encoded as embeddings by a CNN (*e.g.*, ResNet). Contrastive learning trains the CNN to push embeddings from positive pairs closer together while pushing negative pairs farther apart. The resulting CNN encoder generates features that are generally discriminative in the domain of the training images.

### Supervised fine-tuning

Fine-tuning was performed using 8-fold cross-validation. Splits were stratified by diagnosis, age, sex, and visual acuity. CNN weights were fixed during fine-tuning, with only the MLP tuned using a binary cross-entropy loss function, with the two classes specified as control and FTMH. The best model per split was selected by the validation epoch with minimum validation loss divided by validation accuracy. The 8-fold split model outputs were combined as an averaged ensemble with equal weights between components. Fine-tuning data included up to three B-scans per eye centered on the fovea.

### ImageNet pre-trained model

A ResNet50 model pretrained on ImageNet was loaded from the Keras library with weights kept trainable during fine-tuning. A MLP head was attached, comprised of three fully connected layers with 512 nodes per layer. Fine-tuning was performed and evaluated using 8-fold cross-validation as described above.

### Challenge data set

After the model was trained to distinguish FTMH from control (*i.e.*, ERM), the CNN was challenged by OCT scans from a separate patient group who met the inclusion and exclusion criteria and were diagnosed with pseudohole/lamellar holes (n = 34). This diagnosis was confirmed by the same three trained readers who distinguished the FTMH and ERM diagnosis in the pre-training dataset. The central B-scan for each patient was determined by the readers, and the central B-scans were classified by both the ImageNet and RaSCL models as FTMH positive or negative. The challenge dataset was used to assess the generalization of the model to an unseen condition and thus, patient demographics were not considered.

## Results

### Patient characteristics

The working dataset contains OCT images of 61 eyes from 60 patients with FTMH (46 Females, 15 Males, ages 52–84 years, mean: 69.6 years, SD: 6.4 years). The remainder of FTMH-negative control data consisted of scans from patients diagnosed with ERM, comprising 274 eyes from 264 patients (140 female, 134 male, ages 23–93 years, mean: 70.5 years, SD: 8.6 years).

A summary of baseline demographic characteristics, lens status and IOP at time of image acquisition for FTMH and ERM patients is available in Table 1. Each characteristic is well distributed within each group with no significant difference. The only significance was seen for the female population of the FTMH eyes ($p < 0.05$).

### Performance

Ensemble models were evaluated on the holdout test set comprising 10 FTMH B-scans and 40 control B-scans, performed in triplicate for different train-test replicates (Fig 2, Table 2).

### Gradient visualization

Visualization of gradient activation for an FTMH input (Fig 3A) shows that for the RaSCL model, strong activation is localized around the macular hole (Fig 3A'). In contrast, the ImageNet model presents a weaker saliency around the middle of the B-scan (Fig 3A"). Gradient activation for a control input (Fig 3B) shows that the RaSCL model highlights the inner

**Table 1. Patient Demographics for FTMH and ERM eyes.**

| | | Overall | p-value | FTMH n | FTMH p-value | ERM n | ERM p-value | Test of Significance |
|---|---|---|---|---|---|---|---|---|
| Total Number of Eyes | | 334 | | 61 | | 273 | | |
| Age | | 70.3 | > 0.05* | 69.6 | | 70.4 | | T-test |
| Sex | Male | 148 | | 15 | < 0.05‡ | 133 | > 0.05* | Binomial |
| | Female | 186 | | 46 | | 140 | | |
| Eye | OD | 159 | | 30 | > 0.05* | 131 | > 0.05* | Binomial |
| | OS | 173 | | 31 | | 142 | | |
| Lens Status | Phakic | 161 | | 35 | > 0.05* | 126 | > 0.05* | Binomial |
| | Pseudophakic | 172 | | 26 | | 147 | | |
| IOP[a] | | 14.7 | > 0.05* | 14.3 | | 14.8 | | T-test |

\* No statistically significant difference

‡ Statistically significant difference

a Measurement taken at time of optical coherence tomography image acquisition

limiting membrane on the upper surface of the macula, and the undisrupted retinal pigment epithelium below (Fig 3B'). ImageNet similarly presents weak saliency for this scan (Fig 3B").

## Challenge images

A challenge dataset was assembled comprising 34 images presenting lamellar holes, which are partial-thickness defects that can appear similar to FTMH, but do not extend through all the neuronal layers to the RPE. Most lamellar holes do not require intervention. Pseudoholes are another example of a visually similar diagnosis without the need for clinical intervention. We challenged all replicates of each model type and found that all replicates of the RaSCL pre-trained models correctly classified all the challenge data, while the ImageNet pre-trained

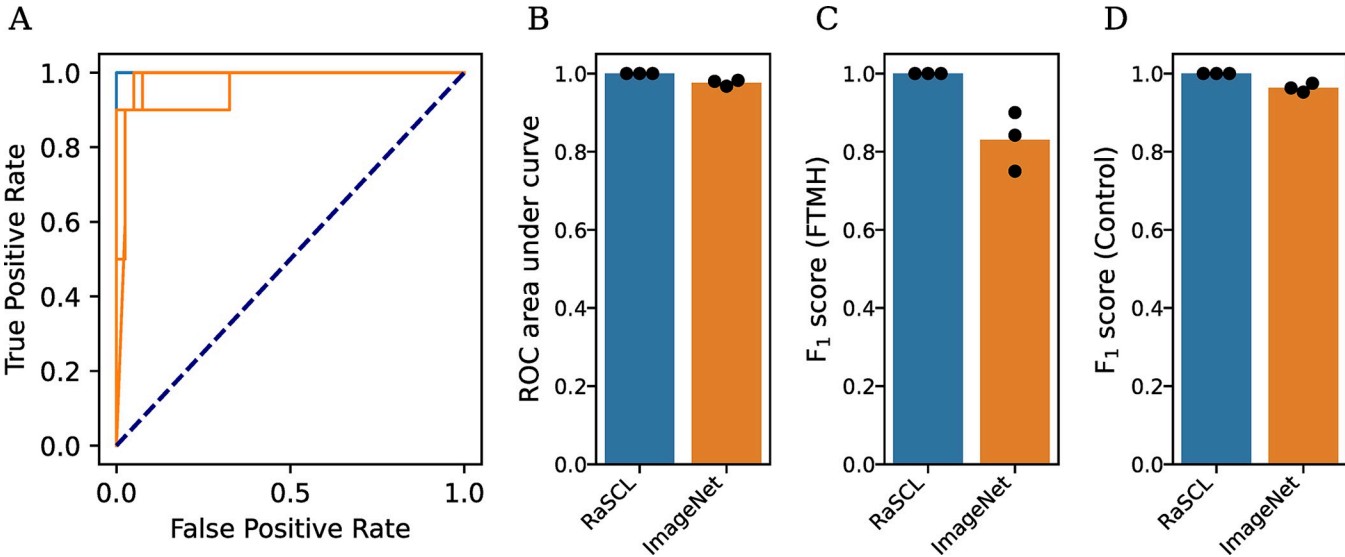

**Fig 2. Summary statistics on three holdout replicates evaluated by models pre-trained with RaSCL or ImageNet.** (A) Receiver operating characteristic (ROC) curves with associated area under curve (B). $F_1$ score for FTMH classification (C). $F_1$ score for control classification (D). Bars indicate mean statistics for each group, and dots indicate individual replicates.

**Table 2. Summary statistics on three holdout replicates evaluated by random slice contrastive learning model (RaSCL) or ImageNet-based control model, scoring for detection of FTMH or control condition.**

| Model | Precision | Specificity | Sensitivity | F1 Score | | Scans | |
|---|---|---|---|---|---|---|---|
| | | | | FTMH | Control | FTMH | Control |
| RaSCL | 1.000 ± 0.000 | 1.000 ± 0.000 | 1.000 ± 0.000 | 1.000 ± 0.000 | 1.000 ± 0.000 | 10 ± 0 | 40 ± 0 |
| ImageNet | 0.930 ± 0.050 | 0.983 ± 0.012 | 0.767 ± 0.125 | 0.831 ± 0.062 | 0.963 ± 0.009 | 10 ± 0 | 40 ± 0 |

The 3D spatial contrast pre-trained network performed completely accurately, diagnosing all test scans correctly in all replicates. The ImageNet pre-trained model performed significantly worse, in terms of both receiver operating characteristic accuracy ($p = 0.007$) and $F_1$ score for FTMH classification ($p = 0.018$). Test samples were classified as FTMH or control using a default softmax score threshold of 0.5.

models misclassified several images, with a mean accuracy of 0.971, $p = 0.158$. We visualized the FTMH gradients to explore possible explanatory factors for the misclassifications (Fig 4B–4D). While RaSCL FTMH gradients had strong activation around the pseudohole (Fig 4C), this activation did not extend down to the RPE and did not lead to misclassification. By contrast, ImageNet FTMH gradients were diffused around the tissue, suggesting poor saliency from the domain transfer model (Fig 4D).

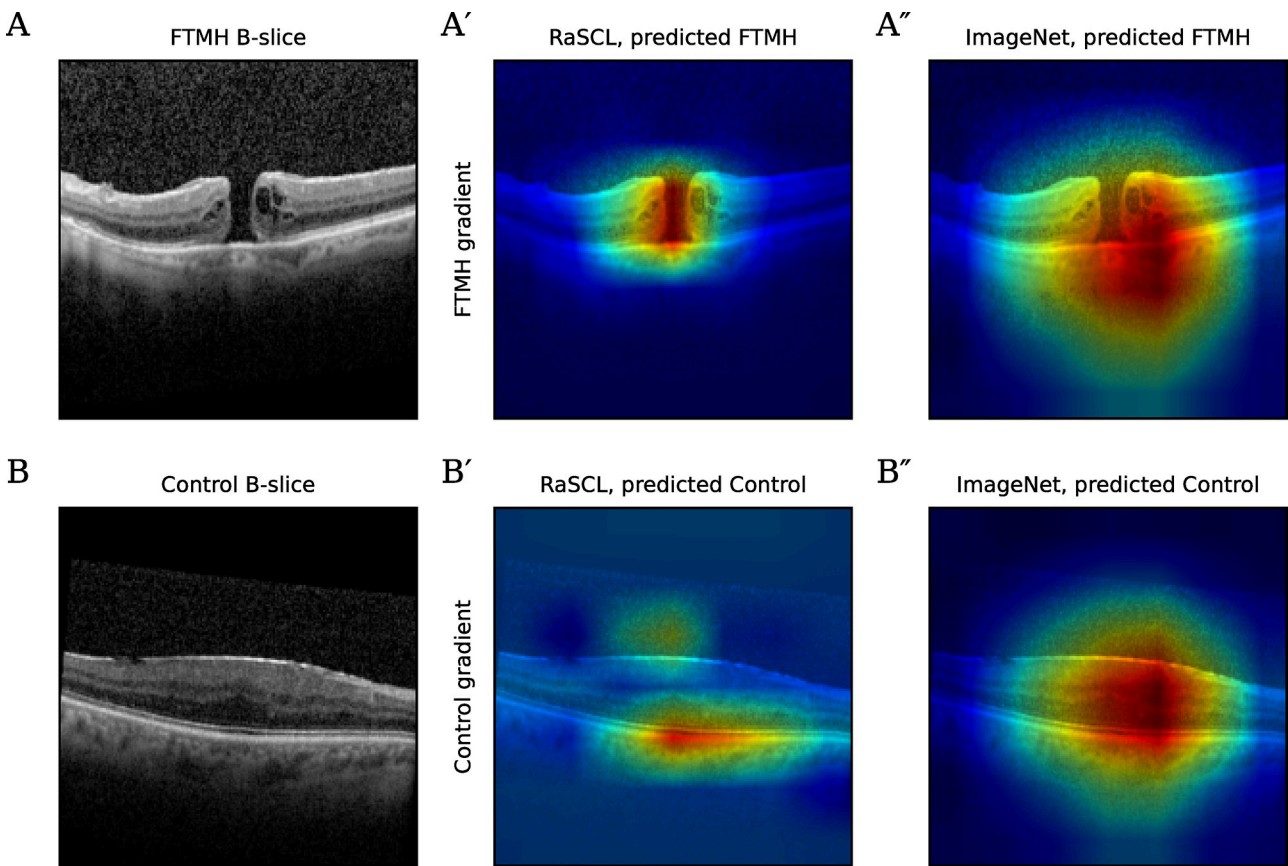

**Fig 3. Gradient visualizations of FTMH and control B-scan images for RaSCL and ImageNet pre-trained models.** (A) Gradient visualization of the FTMH output unit using an FTMH B-scan. (B) Gradient visualization of the control output unit using a control B-scan.

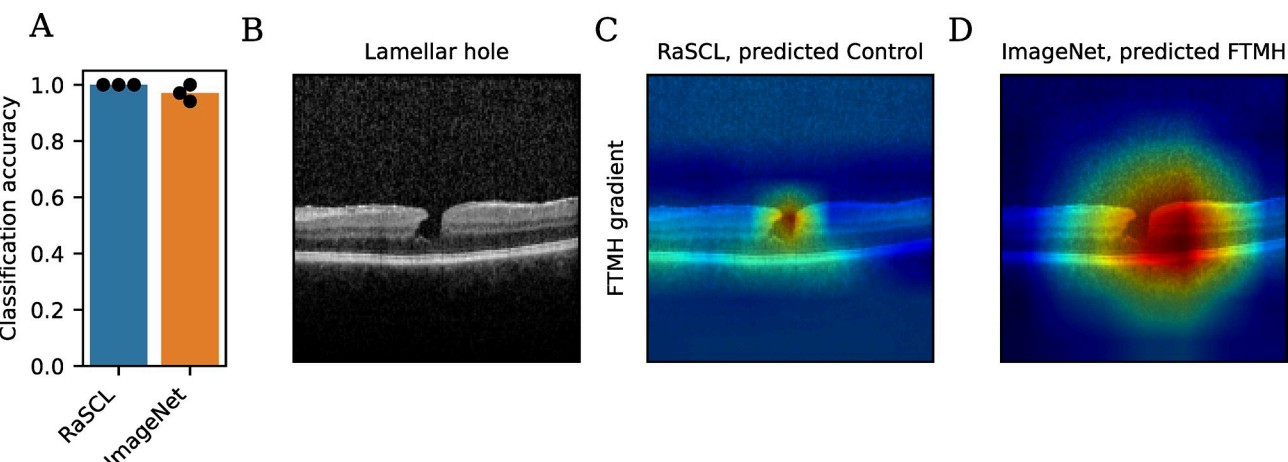

**Fig 4. Challenge image set performance.** (A) Classification accuracy. Bars indicate mean and dots indicate replicates. (B) Challenge image with a lamellar hole. (C) Image correctly classified by the RaSCL model overlaid with FTMH gradients. (D) Image incorrectly classified by ImageNet pre-trained models with overlaid FTMH gradients.

## Discussion

The use of artificial intelligence (AI) in ophthalmology has advanced significantly, owing to the many image-based investigations in the field. This work demonstrates a potentially strong diagnostic model for full-thickness macular holes, which are a serious eye care concern due the significant vision impairment resulting from the characteristic defect in the central fovea and subsequent loss of central visual acuity [4]. Previous attempts on automated FTMH classification have tended to use supervised ImageNet pre-trained networks with good results but which leave room for improvement. The present work demonstrates that near perfect accuracy is achievable for FTMH classification, although the small size of the dataset justifies only modest conclusions about model generalizability on larger datasets.

There are three options when attempting to train deep learning models on small datasets. (*i*) The first option is transfer learning: to pre-train the model on data from an unrelated domain, typically ImageNet. We have shown that this produces models with inferior performance and ambiguous interpretability. (*ii*) The second option is self-supervised learning: to pre-train the model using self-derived labels and contrastive learning on data from the same domain. This is the method of the present article, which we have shown leads to strong models with good performance and interpretability, even when the dataset used for self-supervised learning is modest in size. (*iii*) The third option is to fine-tune a foundation model that is trained on large and diverse dataset in the domain of interest. A retina-specific foundation model was published very recently [44], at around the same time that the present manuscript was in the final stages of submission. The present authors believe that this third option of fine-tuning a foundation model holds promise, and we leave it to future work to do a comparison between all three options.

The method introduced here, namely random-slice spatial contrastive learning, allows the network to develop a good image representation learned from the data, which provides features that are strongly discriminative for the downstream task of recognizing FTMH. This work demonstrates that strong purpose-specific pre-training is viable for small dataset sizes, and that this method improves performance over transfer learning models. Visualization of gradient activation (Figs 3 and 4) shows sharp saliency in the RaSCL model. The lower discriminative capacity and poor saliency of transfer learning models may also result in brittle classifiers with inferior generalization.

The CNN models in this study present a good starting point for fine-tuning models addressing other OCT-related diagnostic and prognostic questions, in ophthalmology or other fields of inquiry. Prior works addressed whether the outcome of FTMH-corrective surgery can be predicted, with only moderate success [26,39,40]. A natural step would be to use the RaSCL approach to develop models for accurate surgical prognosis, which could have clinical benefit by helping to avoid unnecessary surgeries. This study highlights spatial contrastive learning as a powerful pre-training approach for medical scans, which can be applied to deep learning models for other imaging modalities to enhance their tractability.

Prior to clinical roll-out, further considerations need to be addressed. This data is limited to one retina practice with four retina surgeons with a predominantly Caucasian patient population, restricted to one device type, and has a narrow set of inclusion criteria. Additionally, our data is limited to patients who have undergone ILMx which may introduce selection bias in our dataset as this excludes patients who may have elected against surgical intervention or patients lost to follow up. To develop higher confidence in the model for deployment, other devices (*e.g*., Zeiss Cirrus, and Topcon), a patient population with a broader demographic profile, a balanced sex distribution and disease-agnostic OCT data would be required. Another frequent issue in training deep learning systems with clinical data is low dataset size. Datasets ranging in the low hundreds of patients are common. This makes it impractical to train a naïve network to a specific problem directly and has typically been circumvented by fine tuning a pretrained network. This can be problematic, as the pretraining is usually based on images from a different domain, which means that large portions of the network are irrelevant to the specific nuances of the tissue/imaging method.

We can apply the framework presented here to an expanded dataset to address these limitations. We included challenge data showing performance on lamellar holes (Fig 4) which is a condition that could be conflated with FTMH. Results on the challenge dataset consisting of lamellar/pseudoholes corroborate the robustness of contrastive pre-training to yield improved performance over non-medical domain pre-trained networks.

The findings in this study illustrate that deep learning algorithms can be used for computer-assisted screening of FTMH in optometry and primary care settings, promoting appropriate and timely referrals to retinal specialists. Efficient patient triage streamlines clinical workflow, reduces clinician workload, and expedites referrals so that patients get access to care sooner, improving overall outcomes.

## Author Contributions

**Conceptualization:** Timothy William Wheeler, Andrew Clark Thomson, Allan Hunter, Couroosh Mehanian.

**Data curation:** Timothy William Wheeler, Kaitlyn Hunter, Patricia Anne Garcia, Henry Li, Andrew Clark Thomson, Allan Hunter, Courosh Mehanian.

**Formal analysis:** Timothy William Wheeler, Allan Hunter, Courosh Mehanian.

**Investigation:** Timothy William Wheeler, Kaitlyn Hunter, Patricia Anne Garcia, Courosh Mehanian.

**Methodology:** Timothy William Wheeler, Allan Hunter, Courosh Mehanian.

**Software:** Timothy William Wheeler, Courosh Mehanian.

**Supervision:** Timothy William Wheeler, Allan Hunter, Courosh Mehanian.

**Validation:** Timothy William Wheeler, Courosh Mehanian.

**Writing – original draft:** Timothy William Wheeler, Kaitlyn Hunter, Andrew Clark Thomson, Allan Hunter, Courosh Mehanian.

**Writing – review & editing:** Timothy William Wheeler, Kaitlyn Hunter, Patricia Anne Garcia, Henry Li, Allan Hunter, Courosh Mehanian.

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
