## [Decision Letter · Decision Letter 0]

6 Feb 2024

PDIG-D-23-00420

Self-supervised contrastive learning improves machine learning discrimination of full thickness macular holes from epiretinal membranes in retinal OCT scans

PLOS Digital Health

Dear Dr. Hunter,

Thank you for submitting your manuscript to PLOS Digital Health. After careful consideration, we feel that it has merit but does not fully meet PLOS Digital Health's publication criteria as it currently stands. Therefore, we invite you to submit a revised version of the manuscript that addresses the points raised during the review process.

Please submit your revised manuscript within 60 days Apr 06 2024 11:59PM. If you will need more time than this to complete your revisions, please reply to this message or contact the journal office at digitalhealth@plos.org. Please include the following items when submitting your revised manuscript:

We look forward to receiving your revised manuscript.

Kind regards,

Po-Chih Kuo, Ph. D.

Guest Editor

PLOS Digital Health

Journal Requirements:

Additional Editor Comments (if provided):

I agree with Reviewers' suggestion that the authors share their data/code with the community and provide a detailed description of their methodology.

Reviewers' comments:

Reviewer's Responses to Questions

**Comments to the Author**

1. Does this manuscript meet PLOS Digital Health’s publication criteria? Is the manuscript technically sound, and do the data support the conclusions? The manuscript must describe methodologically and ethically rigorous research with conclusions that are appropriately drawn based on the data presented.

Reviewer #1: Partly

Reviewer #2: Yes

2. Has the statistical analysis been performed appropriately and rigorously?

Reviewer #1: N/A

Reviewer #2: Yes

3. Have the authors made all data underlying the findings in their manuscript fully available (please refer to the Data Availability Statement at the start of the manuscript PDF file)?

Reviewer #1: No

Reviewer #2: No

4. Is the manuscript presented in an intelligible fashion and written in standard English?

Reviewer #1: Yes

Reviewer #2: Yes

5. Review Comments to the Author

Reviewer #1: Review of "Self-supervised contrastive learning improves machine learning discrimination of full-thickness macular holes from epiretinal membranes in retinal OCT scans." The authors present a promising approach to enhance small datasets through self-supervised contrastive learning.

* Introduction:

 * Incorporate references in the introduction section to support the statements made. Several paragraphs lack proper citations.

 * Clarify the rationale for specifically comparing ERM to full-FTMH. While ERM can lead to macular edema and retinal anatomy distortion, it is not clinically similar to FTMH. Discuss why this specific comparison was chosen.

 * Explore the clinical generalizability of the findings. Explain whether it makes clinical sense for an automated system to distinguish between ERM and FTMH, and why this distinction is relevant in practice.

* Patient Data:

 * Clearly describe the patient data selection process and its source. Emphasize any potential selection bias in the FTMH and ERM datasets, particularly since the data is derived from patients who underwent ocular surgery.

 * Specify whether the research group excluded other macular pathologies besides diabetic maculopathy, and provide details on any other findings that were considered and present.

 * I suggest adding race and ethnicity description.

 * Were applied any quality control screening for OCT exams?

* FTMH:

 * Specify whether the FTMH considered in this study is limited to idiopathic macular holes. Even though some studies showed no effect on delay, the cited study (ttps://doi.org/10.1016/j.ophtha.2022.08.028) identified worse outcomes in postponed surgeries for idiopathic macular holes.

* Macular holes have different classifications, such as the Gass, where stages 2-4 are full thickness but present differences. Add the Gass or IVTS classification for the included FTMH slabs.

* Model Development:

 * On page 5, when discussing "Developing a functional deep learning model," clarify the definition of a "functional model”.

* Image-Level Description:

 * Include a comprehensive description of image-level findings, specifying the number of abnormal slabs detected in the images. This will provide a clearer understanding of the scope of abnormalities captured by the model.

I strongly suggest sharing the study data and codebase to enable validation and generalizability studies. In the data availability statement the authors state that “All relevant data are within the manuscript and its Supporting Information files”, but there is no supplemental file available, no available image data and metadata, and no code repository.

Reviewer #2: General Overview:

The manuscript investigates the use of contrastive self-supervised learning for classifying full thickness macular holes (FTMH) in OCT data. The approach is commendable for its novel application to ophthalmic imaging; however, the manuscript requires substantial revisions to address issues related to reproducibility, transparency of results, and comparative analysis.

Major Comments:

1. The absence of a data/code sharing statement significantly hampers the study's reproducibility. A clear statement regarding the availability of data and code is essential.

2. Adding a table with F1 scores per class and sample counts would provide a clearer understanding of the model's performance and enhance result transparency.

3. There is a notable lack of detailed methodology for the training and preprocessing of the baseline ImageNet pre-trained model, which is crucial for a fair comparison.

4. Incorporating a figure of the proposed method would aid in understanding the complex methodology and enhance clarity.

Minor Comments:

1. The discussion should not only address the strengths of the paper but also delve deeply into the implications and limitations. The size of the dataset and possible biases due to sex imbalance should be discussed.

2. A more critical comparative analysis between the proposed model and the baseline in the discussion section would strengthen the manuscript.

3. Emphasis on methodological rigor and transparency across all experimental procedures is paramount for the scientific validity of the research.

4. The initial portion of the Discussion section, which provides background information, should be relocated to the Introduction to better set the context for the study.

5. The decision to utilize an ImageNet pre-trained model as baseline, over domain-specific models such as Retfound (https://www.nature.com/articles/s41586-023-06555-x), necessitates a thorough justification.

6. The manuscript's structural organization needs refinement, including the incorporation of a conclusions section.

6. PLOS authors have the option to publish the peer review history of their article (what does this mean?). If published, this will include your full peer review and any attached files.

**Do you want your identity to be public for this peer review?** For information about this choice, including consent withdrawal, please see our Privacy Policy.

Reviewer #1: Yes: Luis Nakayama

Reviewer #2: Yes: David Restrepo

---

## [Decision Letter · Decision Letter 1]

8 Jul 2024

Self-supervised contrastive learning improves machine learning discrimination of full thickness macular holes from epiretinal membranes in retinal OCT scans

PDIG-D-23-00420R1

Dear Dr. Hunter,

We are pleased to inform you that your manuscript 'Self-supervised contrastive learning improves machine learning discrimination of full thickness macular holes from epiretinal membranes in retinal OCT scans' has been provisionally accepted for publication in PLOS Digital Health.

Best regards,

Martin G Frasch

Section Editor

PLOS Digital Health

Reviewer Comments (if any, and for reference):

Reviewer's Responses to Questions

**Comments to the Author**

1. If the authors have adequately addressed your comments raised in a previous round of review and you feel that this manuscript is now acceptable for publication, you may indicate that here to bypass the “Comments to the Author” section, enter your conflict of interest statement in the “Confidential to Editor” section, and submit your "Accept" recommendation.

Reviewer #1: All comments have been addressed

Reviewer #2: All comments have been addressed

2. Does this manuscript meet PLOS Digital Health’s publication criteria? Is the manuscript technically sound, and do the data support the conclusions? The manuscript must describe methodologically and ethically rigorous research with conclusions that are appropriately drawn based on the data presented.

Reviewer #1: Yes

Reviewer #2: Yes

3. Has the statistical analysis been performed appropriately and rigorously?

Reviewer #1: Yes

Reviewer #2: Yes

4. Have the authors made all data underlying the findings in their manuscript fully available (please refer to the Data Availability Statement at the start of the manuscript PDF file)?

Reviewer #1: Yes

Reviewer #2: Yes

5. Is the manuscript presented in an intelligible fashion and written in standard English?

Reviewer #1: Yes

Reviewer #2: Yes

6. Review Comments to the Author

Reviewer #1: All my comments, concerns, and recommendations have been successfully addressed.

Reviewer #2: The authors have taken the time to clarify the comments. The inclusion of new statistics should be recognized, as well as an open code and dataset that promotes reproducibility of the results. The authors also included a more detailed demographic description of the data, allowing the understanding of the data's context to mitigate biases. Clarifications were added on the metrics and dataset split used for the evaluation, allowing for more clarity in the results. Furthermore, despite the inherent limitations of the experiment and the dataset, the authors acknowledged these limitations in the discussions section.

7. PLOS authors have the option to publish the peer review history of their article (what does this mean?). If published, this will include your full peer review and any attached files.

**Do you want your identity to be public for this peer review?** For information about this choice, including consent withdrawal, please see our Privacy Policy.

Reviewer #1: **Yes: **Luis Nakayama

Reviewer #2: **Yes: **David Restrepo
